# DEM–FEM Coupling Simulation of the Transfer Chute Wear with the Dynamic Calibration DEM Parameters

**Fangping Ye** [1], **Yuezhang Qiang** [1], **Weijie Jiang** [1] **and Xiang Fu** [2],*

1   School of Mechanical Engineering, Hubei University of Technology, Wuhan 430068, China;
    yefangping@hbut.edu.cn (F.Y.); 101910093@hbut.edu.cn (Y.Q.); 102010167@hbut.edu.cn (W.J.)
2   School of Railway Locomotive and Vehicle, Wuhan Railway Vocational College of Technology,
    Wuhan 430205, China
*   Correspondence: 20190134@wru.edu.cn

**Abstract:** Transfer chutes for bulk material conveying systems have significant importance in ship loading and unloading and are 'worn' from large mass flow and fast granular material flow conditions. In this investigation, the impact forces of different granular materials on the transfer chute wear process are considered; the DEM–FEM (Discrete Element Method–Finite Element Method) coupling method was used to calculate the wear and the deformation of the transfer chute. The stress–strain and cumulative contact energy from three different granular materials were analyzed under different working conditions. The results show that the wear, stress–strain, and cumulative contact energy of the transfer chute are closely related to the belt speed, the chute inclination angle, and the types of granular materials; the impact force and the stress–strain on the transfer chute achieves maximum value under a 4 m/s belt speed condition; meanwhile, with the increase of belt speed by 0.5 m/s, the wear of the transfer chute increases 25% and the deformation increases 20%; the shape variable, wear area, and normal cumulative contact capacity of the transfer chute are the smallest with a transfer chute inclination angle from 40° to 45°.

**Keywords:** DEM–FEM coupling; transfer chute wear; dynamic calibration; simulation; deformation

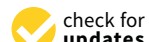



## 1. Introduction

Transfer chutes are widely used in port particle material conveying systems, including the transfer and transportation of particle materials, such as coal mines, iron ore, soybean, and corn. Due to the impact and friction between particle materials and the chute, it often needs to be replaced and repaired, which seriously affects the working efficiency of the whole particle material conveying system [1]. Therefore, how to effectively analyze and reduce the wear, stress, and strain of the transfer chute is particularly important.

In recent years, the discrete element method (DEM) [2] has been widely used in the research of complex particle material flow, and its application scope covers many fields, such as agriculture [3], chemical industry [4], and civil engineering [5]. Before the DEM numerical simulation, the particle DEM parameters must be calibrated; thus, the particle dynamic information can be accurately described and calculated. For example, Sun et al. [6,7] calibrated the coal particle DEM parameters using the static angle of repose, and then the effects of the blade tilt angle on the screw conveying performance were investigated. Liu [8,9] determined the wheat DEM parameters using the static angle of the repose response surface method. However, the research shows that the static experimental calibration DEM parameters are difficult in regard to simulating the dynamic movements of particles. Thus, the dynamic calibration method is considered in this article.

Transfer chutes are often used in loading and unloading particle materials, and their wear is a main concern. Grima [10] evaluated the trajectory of the particle materials, the impact velocity of the particle impact chute, and the resultant force on the transfer chute by DEM simulation. The wear mechanism and wear law of the transfer chute was investigated

by tracking the change of the coal particle position, and it was found that the wear of the scraper conveyor chute increased with the increase of Poisson's ratio, shear modulus, and the density of coal. However, the chute deformation during the wear process was not considered, which affected the location particle impact chute. To improve the research on the wear of the transfer chute, the particle DEM parameters were calibrated by the disc experiment device, and then the DEM–FEM coupling method was used to investigate the details of the transfer chute wear process.

## 2. Numerical Methodology

### 2.1. Discrete Element Method

The particle is described based on the DEM originally proposed by Cundall and Strack [11]; the discontinuous body was separated into a collection of rigid particles; thus, each particle satisfied the motion equation, and the time-step iterative method was used to solve the motion equation of each rigid particle. For particle $i$ at any time $t$, the motion theory can be written as Equations (1) and (2).

$$m_i \frac{d^2 x_i}{dt^2} = m_i g + \sum_{j=1}^{c_i} F_{ij} \tag{1}$$

$$I_i \frac{d\omega_i}{dt} = \sum_{j=1}^{c_i} T_{ij} \tag{2}$$

Here, $m_i$ and $I_i$ are, respectively, the mass and rotational inertia of particle $i$; $x_i$ is the displacement of particle $i$; $\omega_i$ is the angular velocity of particle $i$; $F_{ij}$ and $T_{ij}$ are, respectively, the force and moment of particle $j$ to particle $i$; $c_i$ is the contact number of particle $i$; $g$ is the gravitational acceleration constant, as the DEM particle contact model is shown in Figure 1.

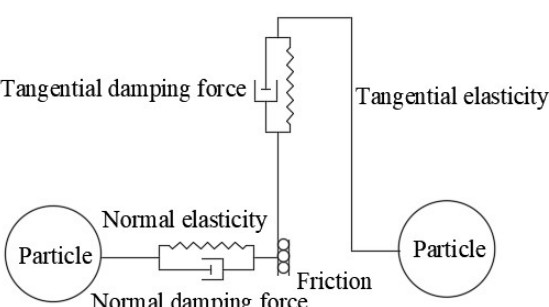

**Figure 1.** Particle contact model.

### 2.2. DEM–FEM Coupling Method

The DEM–FEM coupling method is based on the theory of the surface coupling method [12,13]. The coupling problem is divided into two independent regions—discrete element region and finite element region, respectively. DEM is used to calculate the transfer chute wear process and record the transfer chute load data, and FEM is used to calculate the stress and strain of the transfer chute. Coupling is realized by the iteration of time steps, and the data exchange between the FEM and the DEM are dependent on the coupling interface in each iteration. Specifically, the deformation variables of the chute obtained from FEM are imported into DEM, and then the DEM boundary conditions are updated, and the load data of the transfer chute obtained from DEM are used as the input of FEM. After one update, the DEM calculation continues with the next calculation until the convergent solution is obtained. Therefore, the interaction between particles and the transfer chute can be more truly reflected by the DEM–FEM coupling method, the DEM–FEM coupling flow chart, as shown in Figure 2.

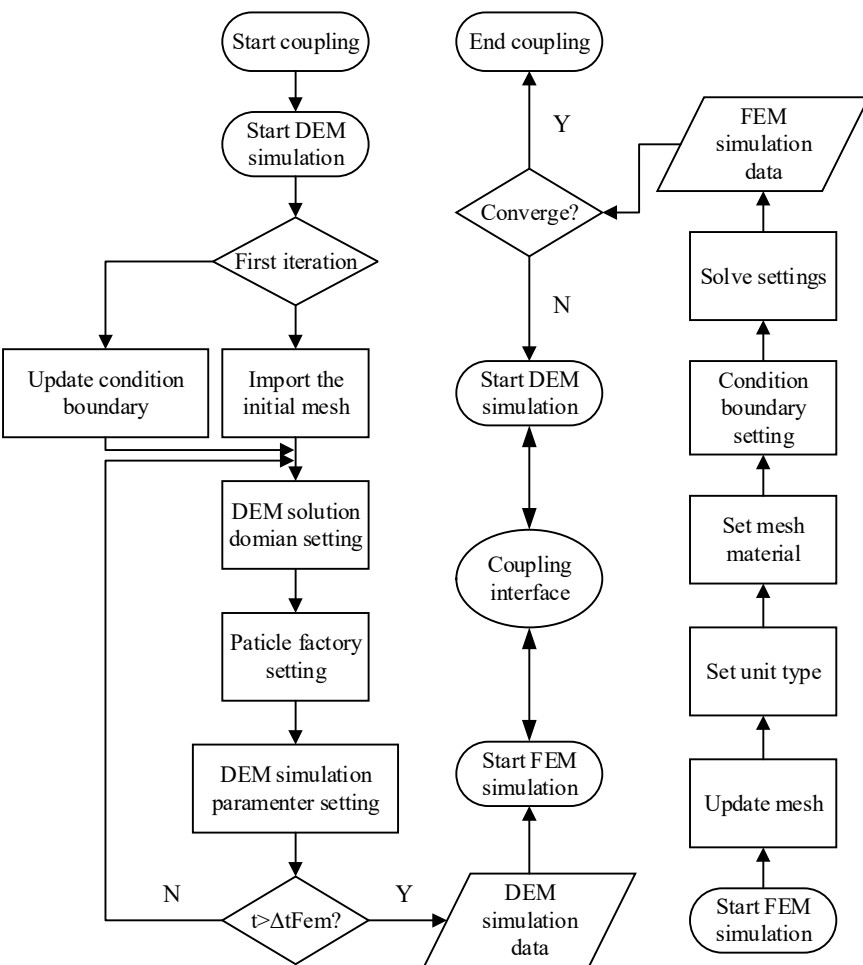

**Figure 2.** DEM–FEM coupling flow chart.

## 3. Simulation Model and Parameters

The simulation model is shown in Figure 3. The particle material are set to flow from the upper hopper to the belt. The relevant physical parameters of the model are shown in Table 1, in which the belt width is 400 mm. Figure 4 shows the particle model of soybean, coal mine, and corn, which is composed of different spherical particles with lengths and diameters of 4, 6, and 5 mm, respectively. Figure 5 shows the distribution of particle materials on the belt, and its section is an arc shape. The relevant physical parameters of the particle are shown in Table 2. $\mu_{s,pp}$: coefficient of static friction between the particle and particle; $\mu_{r,pp}$: coefficient of rolling friction between the particle and particle; $\mu_{s,pg}$: coefficient of static friction between the particle and geometry; $\mu_{r,pg}$: coefficient of rolling friction between the particle and geometry.

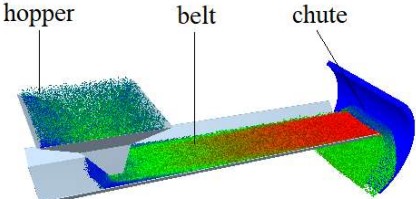

**Figure 3.** Simulation model.

**Table 1.** Parameters of the geometry model.

| Geometry | Hopper | Belt | Chute |
|---|---|---|---|
| Poisson's ratio | 0.31 | 0.49 | 0.31 |
| Shear modulus (Pa) | $1 \times 10^6$ | $1 \times 10^6$ | $1 \times 10^6$ |
| Solids density (kg/m$^3$) | 7800 | 2000 | 7800 |

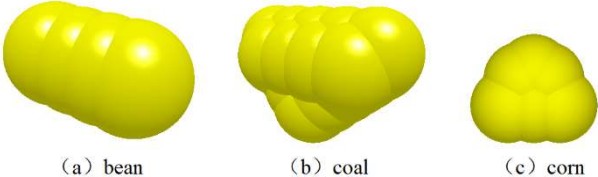

(a) bean      (b) coal      (c) corn

**Figure 4.** Particle model.

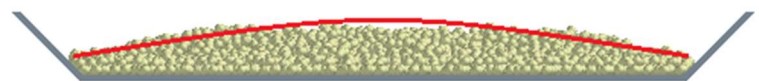

**Figure 5.** Distribution map of the particle.

**Table 2.** DEM Parameters.

| Properties | Coal | Bean | Corn |
|---|---|---|---|
| Solids density (kg/m$^3$) | 1500 | 2000 | 780 |
| Poisson's ratio | 0.3 | 0.25 | 0.438 |
| Shear modulus (Pa) | $1 \times 10^6$ | $1 \times 10^6$ | $1 \times 10^6$ |
| Coefficient of restitution | 0.2 | 0.1 | 0.1 |
| Coefficient of static friction $\mu_{s,pp}$ | 0.4~0.8 | 0.3~0.7 | 0.2~0.5 |
| Coefficient of rolling friction $\mu_{r,pp}$ | 0.01~0.04 | 0.01~0.06 | 0.02~0.04 |
| Coefficient of static friction $\mu_{s,pg}$ | 0.3~0.6 | 0.3~0.5 | 0.2~0.5 |
| Coefficient of rolling friction $\mu_{r,pg}$ | 0.01~0.03 | 0.01~0.05 | 0.01~0.02 |

## 4. DEM Parameters Dynamic Calibration

### 4.1. Friction Coefficient Calibration with Disc Tester

To capture the flow state of particles in a dynamic condition, the disc tester was used to calibrate the particle materials, which included measuring rods, tube wall, central axis, horizontal disc, and adjustable speed motor, as shown in Figure 6. The horizontal disc diameter $D_1$ was 190 mm, the gap between the tube and the shell $d$ was 0.4 mm, the vertical distance from the disc surface to the top of the wall $h$ was 168 mm, and the central axis diameter $D_2$ was 20 mm. After the particle materials were put into the device, the horizontal disc was controlled by the adjustable speed motor, and the dynamic free surface of the particle material was formed as the interaction of centrifugal force and friction. The particle material filling rate was 40%, the rotating speed of the disc was set to 200 rpm, and the particle free surface was measured after the particle flow reached a stable state; meanwhile, the DEM simulation model and process were consistent with the disc tester. Figure 7 shows the bean particle free surface morphology; it is clear that the results of the DEM simulation are consistent with the disc experimental results, and Table 3 shows the calibration results.

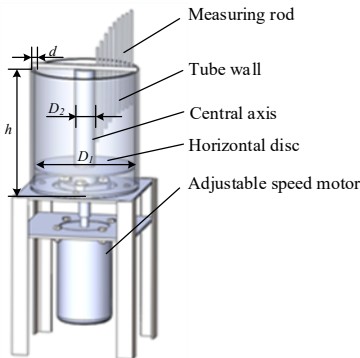

**Figure 6.** Rotating disc device.

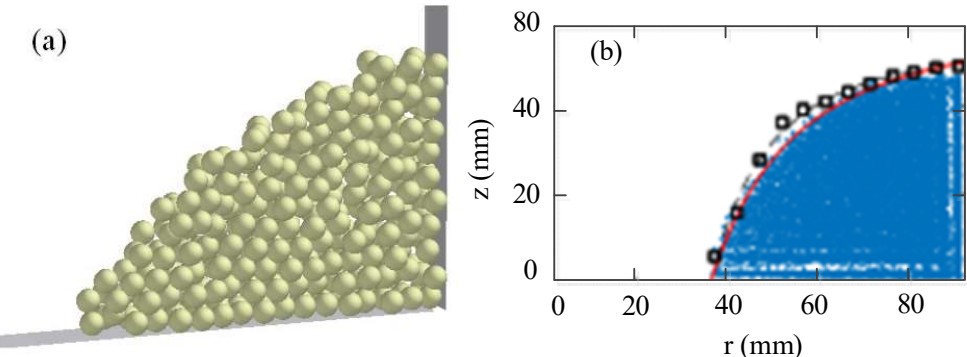

**Figure 7.** Free surface morphology of particles; (**a**) DEM simulation, (**b**) comparisons between simulation and disc tester.

**Table 3.** Calibration results of the DEM Parameters.

| Properties | Coal | Bean | Corn |
|---|---|---|---|
| Coefficient of static friction $\mu_{s,pp}$ | 0.42 | 0.34 | 0.31 |
| Coefficient of rolling friction $\mu_{r,pp}$ | 0.012 | 0.009 | 0.008 |
| Coefficient of static friction $\mu_{s,pg}$ | 0.47 | 0.34 | 0.32 |
| Coefficient of rolling friction $\mu_{r,pg}$ | 0.018 | 0.015 | 0.016 |

*4.2. Wear Coefficient Calibration with the Wear Tester*

The main wear mechanism of the transfer chute is abrasive wear, the Archard wear model [14–16] can be used in the DEM simulation. The wear depth is directly proportional to the friction stroke, normal load, and wear coefficient, and inversely proportional to the hardness of the material. However, it is necessary to determine the wear coefficient *H* using the wear model, which is determined from Equation (3):

$$H = \frac{H_Z}{T_Z} \tag{3}$$

where $H_z$ is the total wear, $T_Z$ is the total time for operating, *H* is the wear depth per hour.

In order to calibrate the wear coefficient *H*, the wear tester was fabricated, which included the motor, coupling, central shaft, disc, dust cover, and material tank. The disc was fixed on the central shaft by bolts, and the motor drove the central shaft through the coupling, to make the disc rotate, as shown in Figure 8. The length of the central shaft was 500 mm, the diameter of the disc was 150 mm, the gap between the disc and the material slot was 3 mm, the spacing between the discs was 80 mm, and the speed of the motor was 2000 rpm. The particle materials were put into the material tank; the distance between the discs could be adjusted by moving the center axis, causing a certain amount of pressure

between the particle and the disc. The wear results of the disc were measured after the disc stopped. The wear test materials were soybean, coal, and corn, respectively. To be consistent with the simulation, the tests were used to calibrate the wear coefficient between each piece of material and the chute.

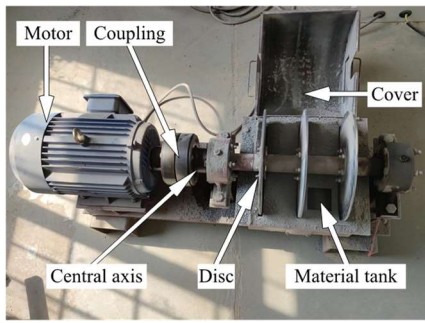

**Figure 8.** Wear test bench.

Figure 9 shows the disc wear results caused by particles in different time periods. The accumulated test time of the disc was 70 days, the change of material was every 2 h during the experimental test. The measured maximum wear amount was 0.079 mm; thus, the wear depth per hour was $4.317 \times 10^{-5}$ mm. Figure 10 shows the DEM simulation disc wear results; the distribution of the wear degree of the disc surface is consistent with that of the disc tester. The wear coefficients among coal, soybean, and corn, and the transfer chute, are shown in Table 4.

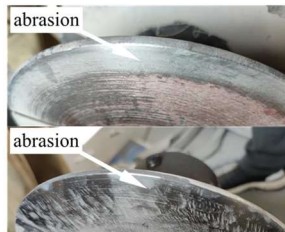

**Figure 9.** Disc wear results.

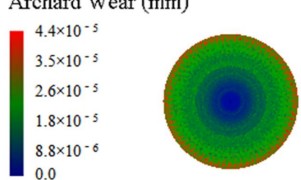

**Figure 10.** DEM simulation disc wear result.

**Table 4.** Coefficient of wear.

| Particle Material | Coal | Bean | Corn |
|---|---|---|---|
| Coefficient of wear | $8.658 \times 10^{-12}$ | $5.364 \times 10^{-12}$ | $6.254 \times 10^{-12}$ |

## 5. Simulation Results

### 5.1. Chute Cumulative Contact Energy

The fundamental cause of wear is the transformation of energy. Energy from friction includes two main parts—one part is dissipated in the form of friction heat; the other part is stored in the chute in the form of elastic potential energy, accounting for about 10% to 15%. The surface of the chute is flaked off in the form of grinding, due to the energy value

of the chute reaching a critical value. Figure 11 shows that the change of the chute contact energy with different simulation times.

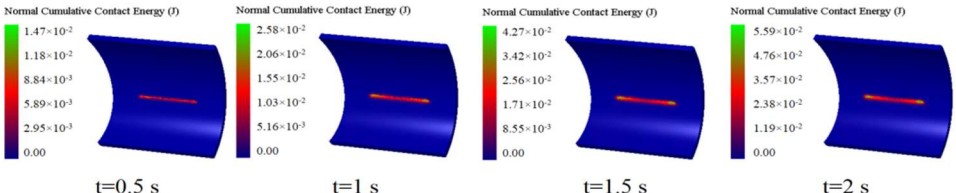

**Figure 11.** Cloud chart of chute contact energy variable change.

As shown in Figure 12, the chute is impacted by particle materials with different angles. As can be seen from Figure 13, the belt speed is proportional to the normal cumulative contact energy. From Figure 13c, the normal cumulative contact energy decreases with the increases in the chute tilt angle when the chute tilt angle reaches a certain value. Due to the increase in belt speed leading to the increase in the speed of particles impacting the chute, many falling particles are blocked by the chute immediately, which increases the wear area between the particles and the chute.

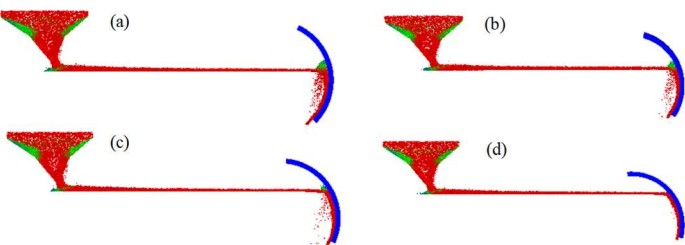

**Figure 12.** Schematic diagram of the chute with different angles (**a**) 10°; (**b**) 20°; (**c**) 30°; (**d**) 40°.

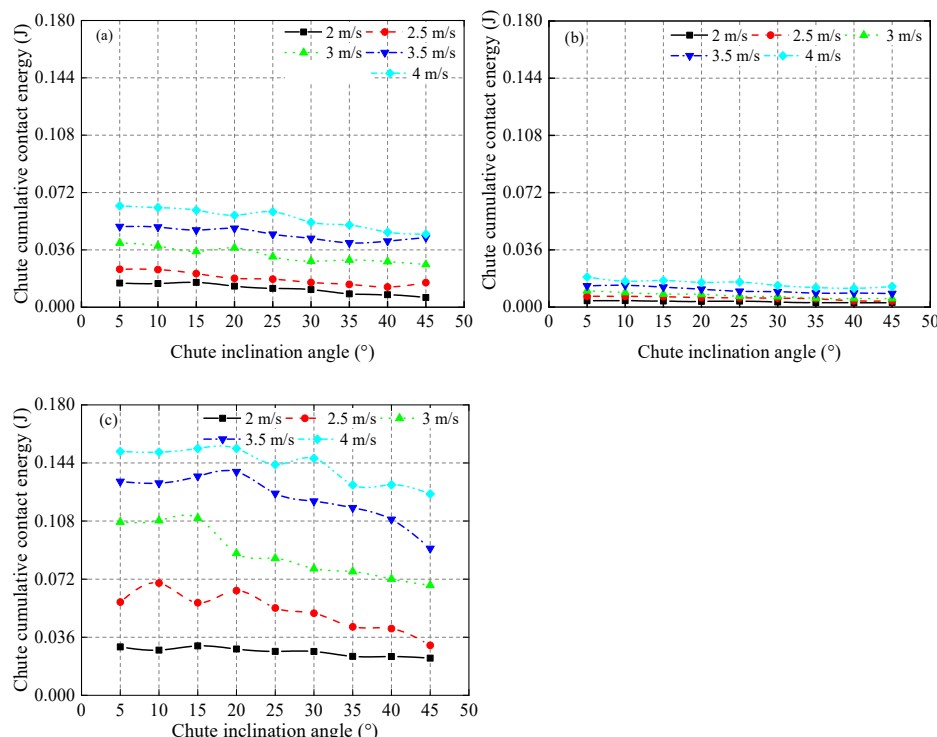

**Figure 13.** Accumulated contact energy by different particles (**a**) coal; (**b**) bean; (**c**) corn.

## 5.2. Chute Wear Volume

The primary cause of chute surface wear is irregular pits and bumps on both the particle material and the chute surface, and the irregularity increases with operating time. Figure 14 presents the simulation diagram of chute wear at different times.

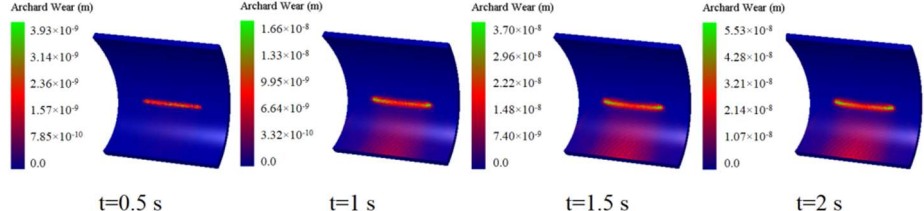

**Figure 14.** Cloud chart of chute wear variable change.

From Figure 15, the change of chute wear is small with the belt speed of 2–3 m/s, but the increase of chute wear is higher under the belt speed of 3.5–4 m/s. With similar chute inclination and belt speed, coal particles have a greater roughness profile, and the chute wear volume is the largest compared to soybean and corn particles.

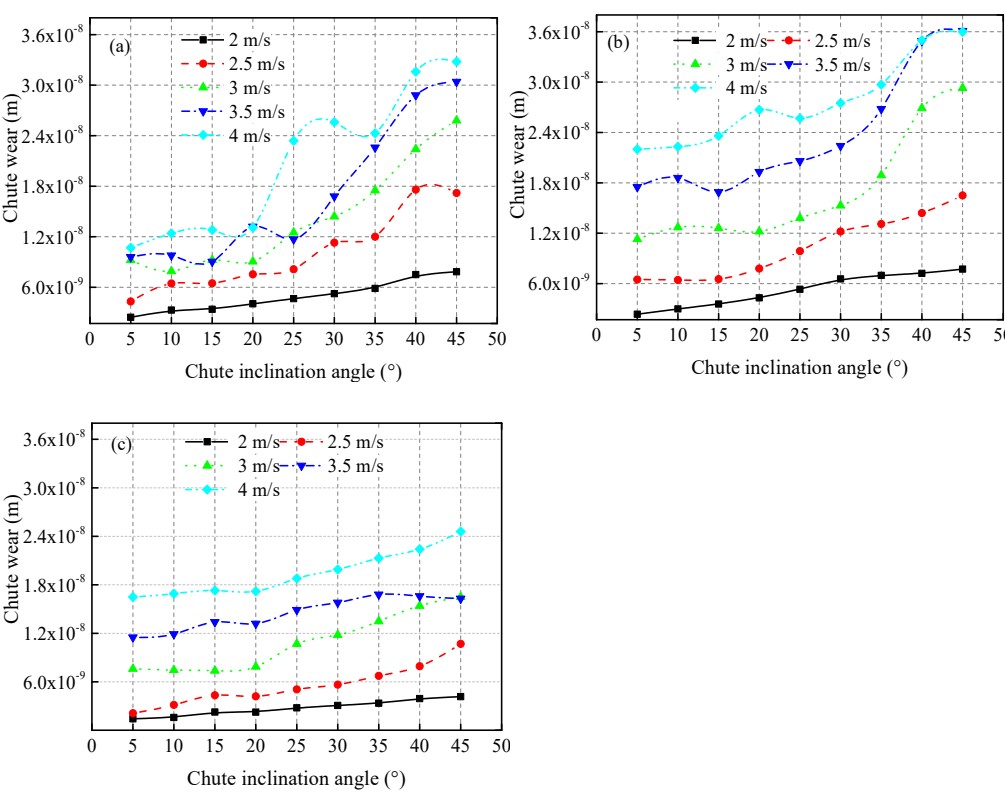

**Figure 15.** Wear volume produced by different particles (**a**) coal; (**b**) bean; (**c**) corn.

## 5.3. Shape Variables

The deformation of the chute is mainly caused by the continuous impact of the particle material on the chute, and the chute deformation is related to the surface shape of the chute and material transfer efficiency. As shown in Figure 16, the maximum deformation of the chute is focused in the middle part of the chute. From Figure 17, the value of the chute deformation variable only fluctuates in a small range, and the change of the chute tilt angle affects the deformation variable by only 6–10%. The deformation of the chute is larger for coal and corn than for soybeans when the chute tilt angle and belt speed are constant. This is due to the fact that the sharp shape of the coal and corn produces a greater force on the chute.

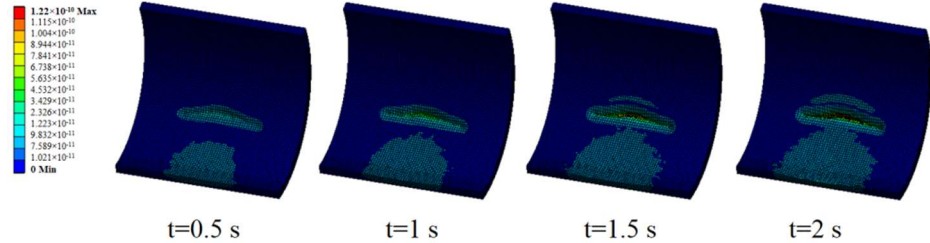

**Figure 16.** Cloud chart of chute shape variable change.

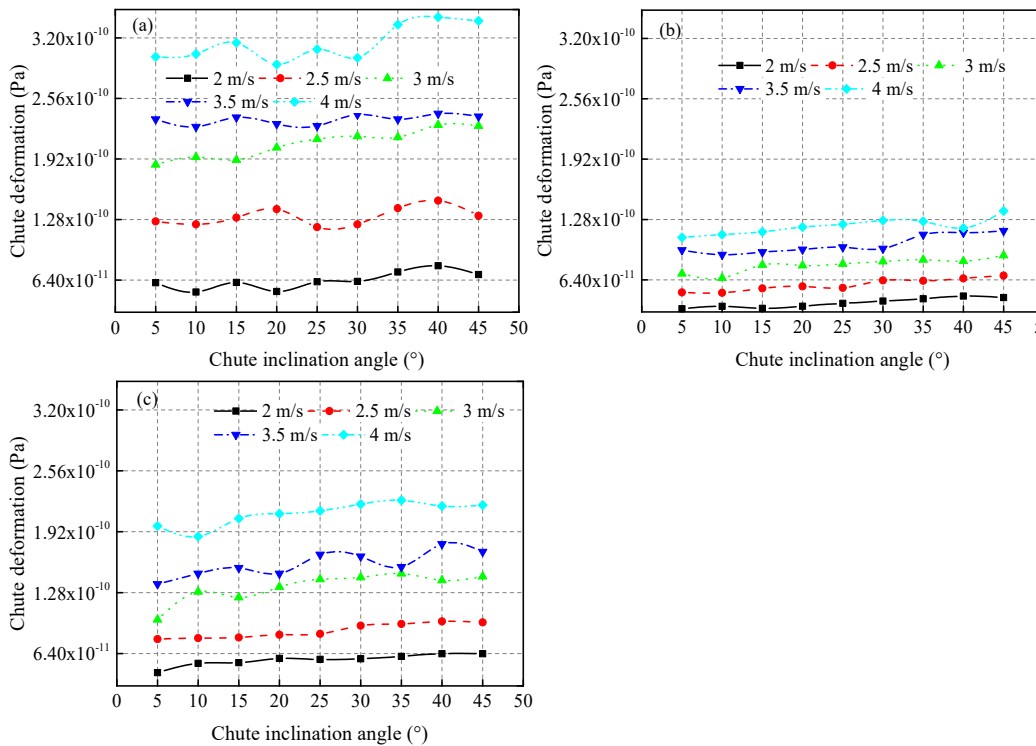

**Figure 17.** Shape variables produced by different particles (**a**) coal; (**b**) bean; (**c**) corn.

### 5.4. Chute Pressure

The main reason for the pressure on the chute is due to the compression between the chute and the particle material, and the value of the pressure on the chute is related to the wear rate of the chute. As shown in Figure 18, the maximum pressure value occurs in the middle of the chute. As shown in Figure 19a, the chute is subjected to a slight change in pressure value at different tilt angles, with a maximum fluctuation of only $0.3 \times 10^3$ Pa. From Figure 19, it appears that at a certain belt speed, the coal particles produce a greater impact on the chute. This is due to the fact that the shape of coal particles is sharper compared to soybean and corn particles, and the smaller contact surface generates more stress on the chute.

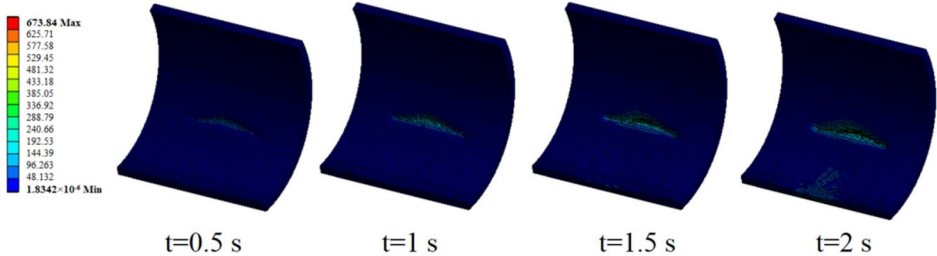

**Figure 18.** Cloud chart of chute pressure value change.

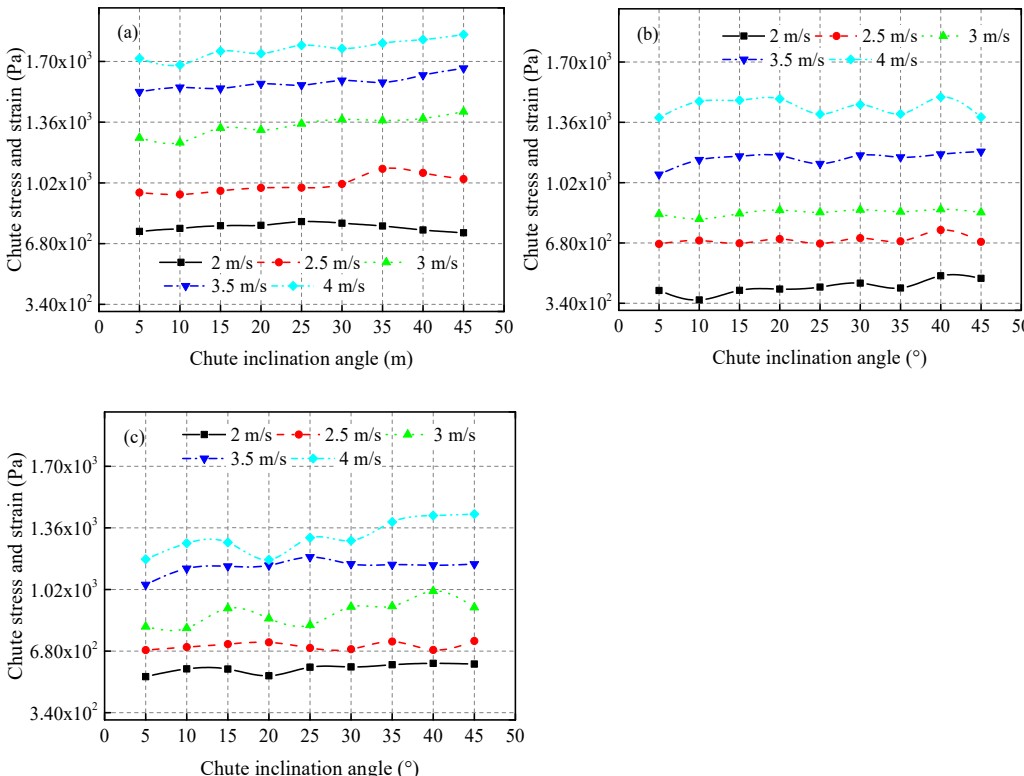

**Figure 19.** Chute pressure produced by different particles (**a**) coal; (**b**) bean; (**c**) corn.

### 5.5. Practical Implications

From the above analysis, the effects of different operating conditions on chute deformation and wear are investigated, which provide a reliable theoretical basis for the actual practical conditions. The belt speed should be selected at 2–2.5 m/s to transport grain pellets and at less than 2 m/s to transport coal pellets. Within the reasonable belt speed range, the effective useful period of the chute can be upgraded.

## 6. Conclusions

The deformation of the transfer chute is predicted by the DEM–FEM coupling method, the impact of coal on the transfer chute wear volume is the largest among the three materials. Furthermore, the changing trend of stress and strain in the coal impact transport chute is the largest with the same belt speed condition.

Among the factors affecting the pressure value and shape variable of the chute, the belt speed is the dominant factor. It can be concluded that the belt speed should be selected as 2–2.5 m/s to convey grain materials and less than 2 m/s to transport coal materials, which provide a reliable theoretical basis for the actual operating conditions.

**Author Contributions:** Conceptualization, F.Y.; methodology, F.Y.; software, Y.Q.; validation, W.J.; formal analysis, X.F.; data curation, Y.Q.; writing—original draft preparation, Y.Q.; writing—review and editing, F.Y. All authors have read and agreed to the published version of the manuscript.

**Funding:** This research received no external funding.

**Institutional Review Board Statement:** Not applicable.

**Informed Consent Statement:** Not applicable.

**Conflicts of Interest:** The authors declare no conflict of interest.

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
