# Peer review of "DEM–FEM Coupling Simulation of the Transfer Chute Wear with the Dynamic Calibration DEM Parameters"

_processes, doi:10.3390/pr9101847_

Round 1
Reviewer 1 Report
Dear Authors, Below is some feedback that may help improve the content of the paper.
- Some improvement of the English writing style is required. Please seek a review from a native speaker, or professional service.
- Section 4.1, line 108 - 110. When describing the disc tester, add the dimensions to Figure 6.
- Why were these materials chosen? They are strikingly different.
- Some more information is required for the wear test in Fig 8.
- Firstly, what is the wear material in the tester, what is its relevance to installed transfer chutes, does it have a practical use for all materials chosen, and is it the same as the material used in the simulations?
- Secondly, these materials have been tested in the wear tester for 70 days. I suspect a significant amount of material degradation would result, impacting the results. The generation of fines and dust due to breakage would result in an invalid result. How was this accounted for, considering a transfer chute would experience a continuous stream of 'fresh' material?
- Thirdly, this tester considers abrasive wear only, yet there is a considerable amount of impact in the simulations. How was this considered? How is differences in normal pressure of the material on the wear surface accounted for between the tester and transfer chute sims?
- Line 157 - discuss the forms of potential energy stored in the chute.
- The design of the transfer chute is not practical, and unrealistic. A standard hood design would be angled to equal the trajectory of the incoming material. Consider running simulations with a real world chute design, or justify your use of this design.
- The design of the chute causes some conclusions to be redundant. The statement that a higher belt velocity causes more wear, for a given chute angle is inconclusive. Of course this is the case, but increasing the belt velocity increases the portion of impact wear compared to abrasive wear, introducing a second mechanism. Both variables (chute angle and velocity) are in fact being changed, so no conclusions can be drawn.
- Line 216 - the statement that an increase in belt velocity increases chute pressure due to particle shape. This is incorrect. As I mentioned earlier, the trajectory of the material changes, and therefore the impact angle.
- Add a section discussing practical implications. How are these materials traditionally handled, what types of chute do they employ, typical belt speeds (relating to chute velocities). What does all of this research "mean"?
- The conclusion needs significant development.
Please consider these revisions.
Reviewer 2 Report
The paper presents an interesting model of the DEM-FEM connection allowing for simulation and research of transport systems. Conveyor transport systems are widely used, which is why the authors' research results may have a wider impact on the technique.
Round 2
Reviewer 1 Report
Thank you for the changes. Please consider the previous recommendations in further detail when designing future research plans.